# The Influence of Tempering Temperature on Retained Austenite and Ductility–Toughness of a High-Strength Low-Carbon Alloyed Steel

**Lirong Sun [1,2,\*], Jiafeng Wan [2], Jiqing Zhang [2], Feng Wang [2], Guo Yuan [1] and Guodong Wang [1]**

[1] State Key Laboratory of Rolling and Automation, Northeastern University, Shenyang 110819, China; yuanguo@ral.neu.edu.cn (G.Y.); wanggd@mail.neu.edu.cn (G.W.)

[2] Shandong Iron and Steel Group Rizhao Co., Ltd., Rizhao 276800, China; wanjf0026@163.com (J.W.); zjq1st@163.com (J.Z.); xiake7@sohu.com (F.W.)

[\*] Correspondence: sunlirong@sdsteelrz.com

**Abstract:** High-strength alloyed steel has been widely used in engineering equipment because of its exceptional strength and toughness, particularly at low temperatures. However, the performance of high-strength alloy steel has not been fully developed, and it is necessary to further optimize the microstructure and mechanical properties. Therefore, the focus of this study is on the phase transition and corresponding mechanical properties of high-strength low-carbon alloyed steels. Three experimental steels were austenitized at 900 °C for 1 h, followed by water quenching, and were then tempered at 570, 600, and 630 °C. They were denoted as QT570, QIT600, and QIT630, respectively. The results show that appropriate intercritical tempering of QIT600 steel significantly increases the proportion of retained austenite and promotes VC precipitation within tempered martensite in comparison to QT570 and QIT630 steels. The enrichment of multiple alloys improved the thermal stability of retained austenite, which was further demonstrated with low-temperature insulation tests. Meanwhile, QIT600 steel with 18 vol.% of retained austenite achieved a superior yield strength of 1025 MPa, an elongation of 21%, and a cryogenic impact energy of 1.25 MJ/m$^2$. The plasticity induced by the transformation of the retained austenite significantly enhanced the strain-hardening rate and postponed necking, thereby increasing elongation. The retained austenite enhanced cryogenic toughness by significantly arresting crack growth and increasing the ability of plastic deformation.

**Keywords:** high-strength steel; retained austenite; intercritical tempering; low-temperature toughness

## 1. Introduction

High-performance structural steel requires high strength, good ductility, excellent low-temperature toughness, and a reasonable yield ratio and has great potential in transportation, energy exploitation, and national defense security applications such as aircraft landing gear, engineering machinery, aerospace, shipbuilding, offshore platforms, naval vessels, and low-temperature pressure-vessel manufacturing [1,2]. Compared with traditional low-carbon low-alloy steel, low-carbon alloy steel exhibits higher strength due to various strengthening mechanisms such as grain refinement, solid solution, and precipitation strengthening. Precipitation strengthening contributes the most significantly to its overall strength among these mechanisms. When adding Cr, Mo, and V micro-alloying elements to steels, they can combine with C or N to form MC and M$_2$C-type carbides which interact with dislocations to improve the strength of steels [3,4].

The yield strength of the Cu-containing low-alloy high-strength steel developed in recent years has increased to above 900 MPa; however, its low-temperature toughness is poor (−20 °C, less than 0.87 MJ/m$^2$), which limits its application in low-temperature environments [5]. Previous studies [6,7] have shown that increasing Ni content (4.5~10%) in low-alloy steel can result in the formation of more retained austenite (10~23%, volume

fraction) by adjusting the heat treatment process. This retained austenite can blunt crack tips and eliminate stress concentration, thereby reducing the ductile–brittle transition temperature of steel and improving its low-temperature toughness. However, an increase in the content of retained austenite results in a softening of the matrix structure and reduces the yield strength of steel to below 700 MPa [8]. Controlling the retained austenite content to achieve an optimal balance between strength and low-temperature toughness is key in developing high-performance structural steel. Additionally, because many key components of engineering structures must withstand complex loads, ultra-high-strength steel needs to appropriately reduce its yield ratio to meet safety design requirements [9,10].

Low-alloy high-strength steel typically undergoes quenching and tempering (QT) or quenching, intercritical annealing, and tempering (QLT) heat treatments in traditional production processes to accurately regulate its microstructure and properties. For instance, HSLA-100 steel is formed with a tempered martensite structure and dispersed carbides through the QT process to achieve ultra-high strength (1000 MPa). However, its impact energy at $-80$ °C was only 0.46 MJ/m$^2$, indicating insufficient low-temperature toughness [11]. Ni-containing low-alloy steel can form stable retained austenite through the QLT process, which significantly improves low-temperature toughness, but the yield strength decreases significantly [12]. Currently, there is insufficient research on the tempering process of low-carbon alloy steel and its impact on the microstructure and properties.

This study examines the microstructural changes in low-carbon alloy steel at various tempering temperatures and explores how these changes affect mechanical properties. The goal is to reach an optimal balance of strength, ductility, and toughness while providing practical guidance for improving production processes.

## 2. Experimental Procedure

The chemical composition of the high-strength steel was 0.1C-0.2Si-8.5Ni+1.5Mn-0.8(Cr+Mo)-0.06V (wt.%), balance Fe. To manufacture the steel, a 200 kg ingot was melted in a vacuum induction furnace and then cast into a 120 mm thick slab. The slab was heated to 1050 °C for 2 h and then hot-rolled to a 15 mm thick plate through seven passes of rolling between 1000 and 980 °C. After that, the plate was cooled with air to room temperature. The $A_{c1}$ and $A_{c3}$ temperatures of the tested steel were determined to be 588 and 740 °C, respectively, using a thermal dilatometer (Formastor-FII). The plates were then reheated to an austenitizing temperature of 900 °C for 1 h and water-cooled to room temperature. Finally, the water-cooled plates were reheated to tempering temperatures of 570, 600, and 630 °C for 1.5 h and air-cooled to room temperature. The heat treatment was based on phase transformation temperature, as shown in Figure 1. The plates were named QT570, QIT600, and QIT630, respectively.

Tensile specimens and Charpy impact specimens were produced from steel plates with varying heat treatment conditions. The tensile samples had a diameter of 5 mm and a gauge length of 25 mm, in accordance with ISO 6892-1:2019. The Charpy impact specimens measured 10 mm × 10 mm × 55 mm, following the guidelines set by ISO148-1:2016. Tensile tests were performed at room temperature (25 °C) using a universal tensile tester with a crosshead speed of 2 mm/min. Charpy V-notch impact tests were conducted at temperatures of $-80$ °C and $-196$ °C on a pendulum impact tester (ZBC2000 MTS) equipped with appropriate instruments. An average of three experiments was calculated for each test condition.

The test steel was analyzed using scanning electron microscopy (SEM) after being polished and etched in a 4% nitric acid solution. The EBSD samples were electropolished in a 10% perchloric acid solution at 20 V for 20 s to alleviate surface stress, followed by observation using an acceleration voltage of 20 kV and a scan step of 0.2 μm. The AztecCrystal software was used to process the experimental data. TEM samples were subjected to mechanical grinding to 40 μm thickness then double-jet thinning at $-25$ °C with 5% perchloric acid at 45 V operating voltage. X-ray spectroscopy (EDX) was used to analyze the composition of the different phases.

The volume fraction of retained austenite was determined at room temperature using an X'Pert PRO X-ray diffractometer with Co Kα radiation. The experimental parameters included a scanning range of 2θ from 45 to 120°, a step size of 0.02°, a scanning rate of 2°/min, and a voltage of 40 kV with a current of 40 mA. The retained austenite volume fraction was quantitatively analyzed in accordance with the guidelines specified in ASTM E975-13 [13]. The calculation of the volume fraction was based on the integrated intensities of the $(200)\gamma$, $(220)\gamma$, $(311)\gamma$, $(200)\alpha$, and $(211)\alpha$ diffraction peaks. Additionally, the average carbon concentration of the retained austenite ($C\gamma$, in wt.%) was determined using Equation (1) [14]:

$$C_\gamma = (\alpha_\gamma - 3.547)/0.046 \tag{1}$$

where the lattice parameter $\alpha_\gamma$ was determined by measuring the diffraction peaks $(200)_\gamma$ and $(220)_\gamma$.

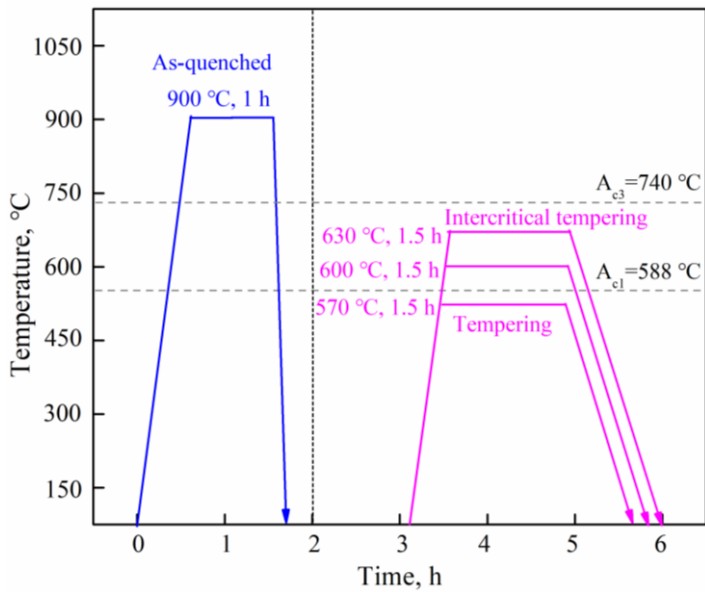

**Figure 1.** Diagram of heat treatment procedures.

## 3. Results and Discussion

### 3.1. Microstructural Analysis

Figure 2 shows SEM images of different heat treatments applied to tested steels. In Figure 2a, a quenched steel with a typical martensite is displayed. The martensite is characterized by a distinct parallel lath structure within each block and the absence of retained austenite. In Figure 2b, the QT570 steel exhibits tempered martensite with numerous martensite–austenite (M–A) constituents. The M–A constituents are located mainly along the boundaries of tempered martensite laths and prior austenite grain boundaries. However, the M–A components disappear with increasing tempering temperature and are thus absent in QIT600 (Figure 2c) and QIT630 (Figure 2d). In Figure 2c, the QIT600 steel displays block-like and strip-like retained austenite, which is high in solute elements. The retained austenite is formed during intercritical tempering and contains a significant amount of solute enrichment. In contrast, the microstructure of QIT630 steel mainly consists of tempered martensite and fresh martensite without any retained austenite present, as shown in Figure 2d. At an intercritical temperature of 630 °C, the reversed austenite increases in size and volume fraction, which in turn decreases its thermal stability and internal element enrichment. This causes most of the reversed austenite to turn into fresh martensite, which has a lath-like morphology in QIT630 steel. The fresh martensite then forms a network structure with tempered martensite laths.

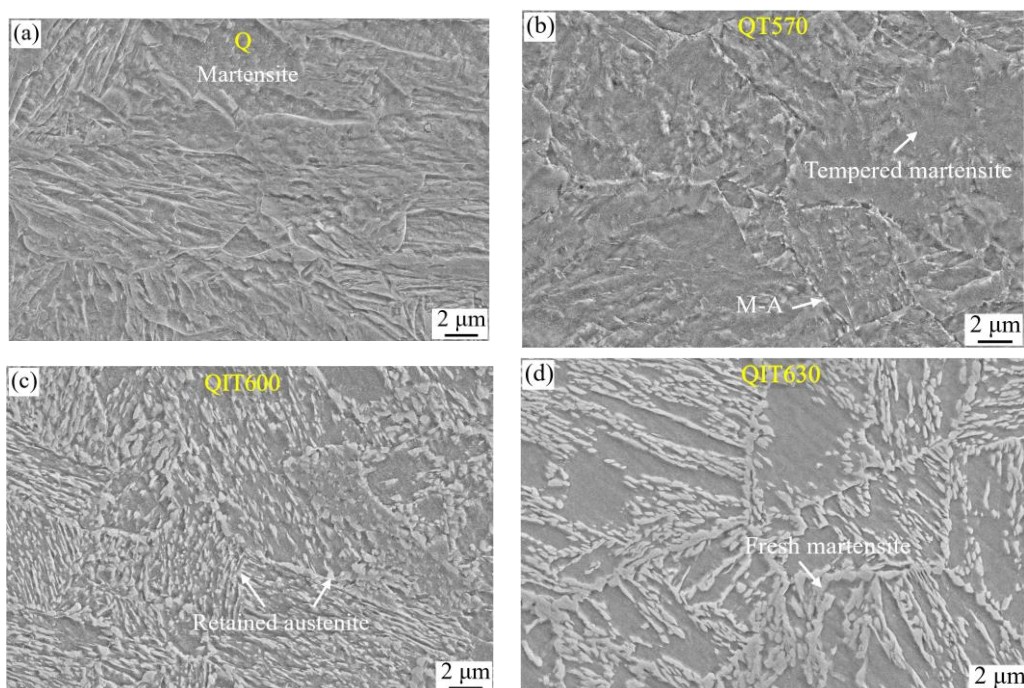

**Figure 2.** SEM micrographs under different heat treatments: (**a**) quenched steel; (**b**) QT570 steel; (**c**) QIT600 steel; (**d**) QIT630 steel.

Using the XRD analysis shown in Figure 3, the volume fraction of retained austenite was determined. It was found that the quenched steel and QT570 steel have no retained austenite at room temperature. However, QIT600 and QIT630 steels have 18 vol.% and 1.5 vol.% retained austenite, respectively (Figure 3a). QT570 steel cannot form retained austenite because of its tempering temperature being below $A_{c1}$ at 570 °C. On the contrary, even after being cryogenically treated at temperatures of −80° and −196 °C for 30 min, the fraction of retained austenite in QIT600 did not decrease. This indicates that the retained austenite in QIT600 steel maintains good thermal stability at low temperatures.

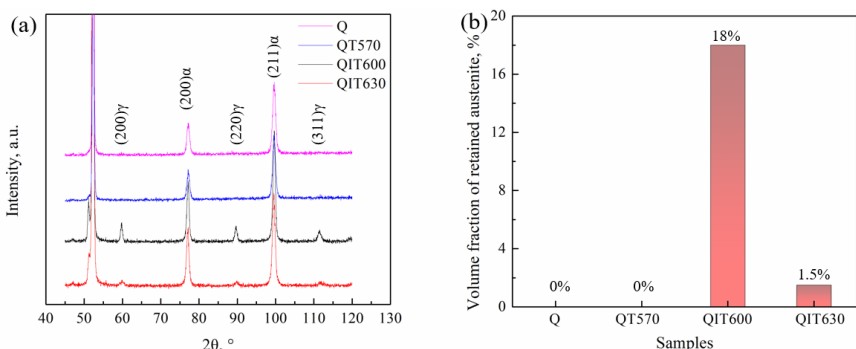

**Figure 3.** XRD spectra: (**a**) retained austenite volume fraction; (**b**) different heat treatments.

Figure 4 presents TEM micrographs that illustrate the changes in matrix structure and precipitates during the tempering process. In the QT570 steel, the microstructures consist of tempered martensite laths and high dislocation density regions within sub-grains (Figure 4a). The presence of Fe-rich $M_3C$-type carbides with a block-like distribution along the grain boundaries is revealed in Figure 4d. The polygonal and massive shape of these carbides can have a negative impact on the ductility and toughness of steel. On the other hand, QIT600 steel showed signs of recovery, such as wider lath widths and fewer dislocations (Figure 4b). The average width of the tempered martensite laths for QT570 and QIT600 steels is 150 nm and 200 nm, respectively. Moreover, QIT600 steel had a higher

density of VC precipitates within the martensite laths (Figure 4e). The VC precipitates were analyzed, and over 200 were found in the test steels. The precipitates in QIT600 steel are spherical with a diameter of 10 nm. In QIT630 steel, the TEM microstructure showed tempered and fresh martensite without any retained austenite (Figure 4c), and the diameter of the largest precipitates increased to 30 nm (Figure 4f). This dual-phase microstructure had a lamellar structure, with the lath width of tempered martensite and fresh martensite measuring 240 nm.

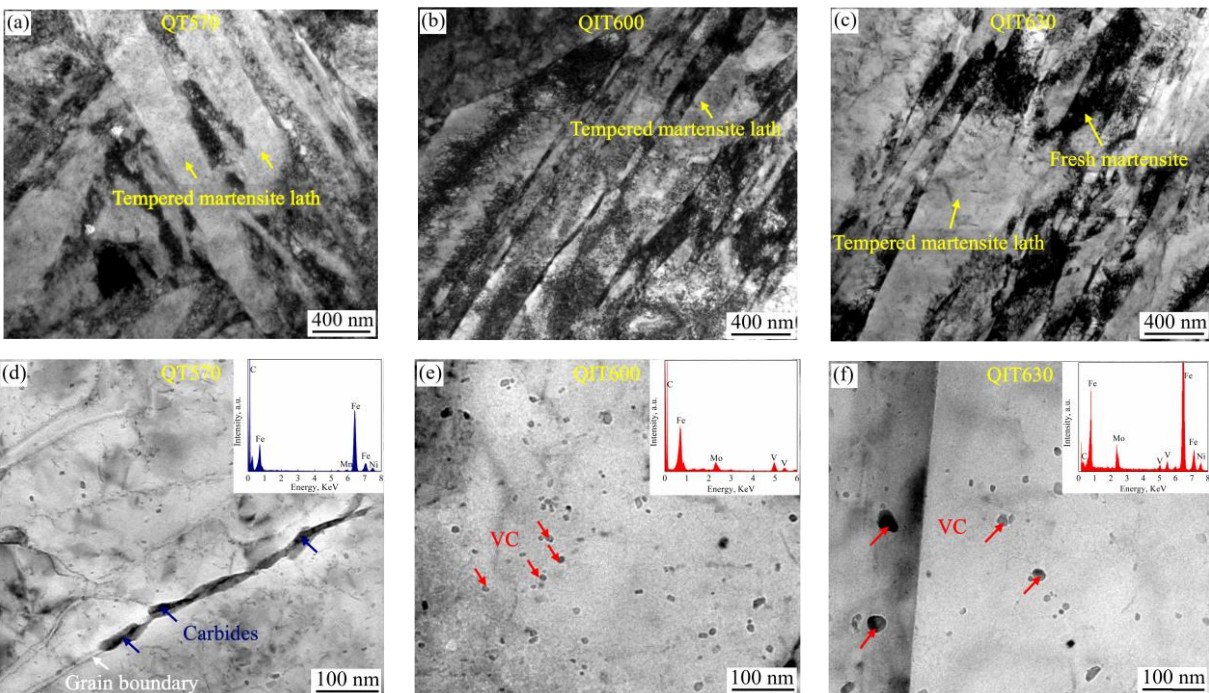

**Figure 4.** TEM micrographs after different heat treatments: (**a**,**d**) QT570 steel; (**b**,**e**) QIT600 steel; (**c**,**f**) QIT630 steel.

The morphology and location of the retained austenite in QIT600 steel were studied using TEM, and the results are shown in Figure 5. Two types of retained austenite were identified: block-like and strip-like. Block-like austenite was found to be discontinuous along the grain boundaries (Figure 5a,b), while strip-like austenite was present between the tempered martensite lath boundaries (Figure 5d,e). The average width of block-like retained austenite was 160 nm, and that of strip-like retained austenite was 50 nm. Figure 5c,f show the chemical characteristics of retained austenite, which were determined by measuring the content of selected austenite stabilizer elements Ni, Mn, and Cr using TEM–EDX. The alloy composition of the block-like retained austenite was 12.15 wt.% Ni, 4.70 wt.% Mn, and 1.96 wt.% Cr, while that of the strip-like retained austenite was 13.64Ni-4.84Mn-2.44Cr wt.%. The strip-like type of retained austenite has higher levels of Ni, Mn, and Cr than the block-like type, which can enhance its stability, as these elements are potent stabilizers of austenite.

The thermal stability of retained austenite is often assessed using the $M_s$ value [15–17]. Earlier research [18,19] has indicated that the influence of elemental composition on $M_s$ can be quantified using Equation (2):

$$M_s(°C) = 539 − 423[C] − 17.7[Ni] − 30.4[Mn] − 12.1[Cr] \tag{2}$$

where [C], [Ni], [Mn], and [Cr] represent the concentration (wt.%) of C, Ni, Mn, and Cr in retained austenite, respectively. The carbon concentration of retained austenite, $C\gamma$ (in wt.%), can be estimated using Equation (1). Applying this equation to the QIT600 steel yielded an estimated C concentration of 0.85 wt.% for its retained austenite. Therefore, the

$M_s$ of block-like and strip-like retained austenite in QIT600 steel was calculated to be $-202$ and $-239\,^{\circ}$C, respectively. The findings suggest that the retained austenite in QIT600 steel demonstrates remarkable thermal stability at cryogenic temperatures, which is consistent with the XRD results shown in Figure 3.

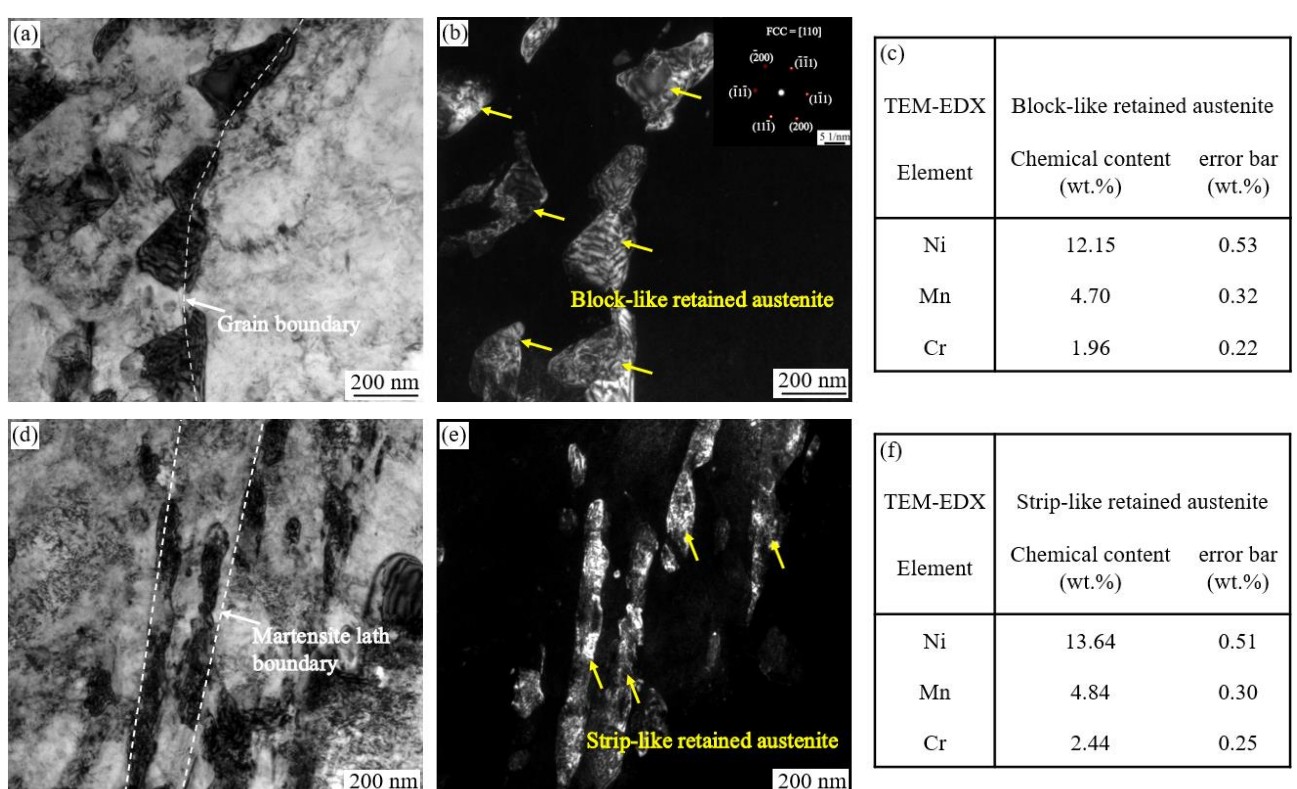

**Figure 5.** TEM micrographs and corresponding chemical content statistics from EDX in the QIT600 steel: (**a**–**c**) block-like retained austenite; (**d**–**f**) strip-like retained austenite.

As shown in Figure 6, EBSD crystal analyses were performed on the QT570, QIT600, and QIT630 steels. The black lines in the figure indicate high-angle grain boundaries (HAGBs) with a misorientation of over 15°, representing block, packet, or prior austenite boundaries. As shown in Figure 6a,c,e, the average block size of tempered martensite gradually increases with higher tempering temperatures because of the merging of blocks caused by the recovery of the tempered martensite matrix. No retained austenite was detected in the tempered martensite matrix of QT570, which is consistent with the XRD results (Figure 3) that show no diffraction peaks of austenite. For QIT600, retained austenite is primarily formed along the HAGBs and within blocks, with a corresponding content of 18 vol.% as determined with XRD. The retained austenite grains within a single prior austenite grain demonstrate consistent IPF color images. Additionally, Figure 6e shows that QIT630 steel has minimal or almost no retained austenite present, with only 1.5 vol.% detected, which is consistent with the XRD results. The region with the dark gray color in Figure 6e represents fresh martensite, while the light gray region is tempered martensite. The QIT630 steel contains a relatively higher amount of fresh martensite due to insufficient martensitic transformation at 630 °C (tempering temperature exceeding $A_{c1}$), which results in unstable reversed austenite transforming into fresh martensite.

The kernel average misorientation (KAM) mappings of QT570, QIT600, and QIT630 are visualized in Figure 6b,d,f, respectively. These maps depict the local orientation gradients within the grain interior. A high KAM value indicates the presence of elevated internal stress and a higher density of dislocations. The KAM maps illustrate that the QIT630 steel exhibits higher KAM values compared to the QT570 and QIT600 steels. This higher KAM

value in QIT630 steel may be attributed to the presence of increased fresh martensite, as observed in Figure 6f, which distinguishes it from the misorientation patterns observed in the QT570 and QIT600 steels (Figure 6b,d).

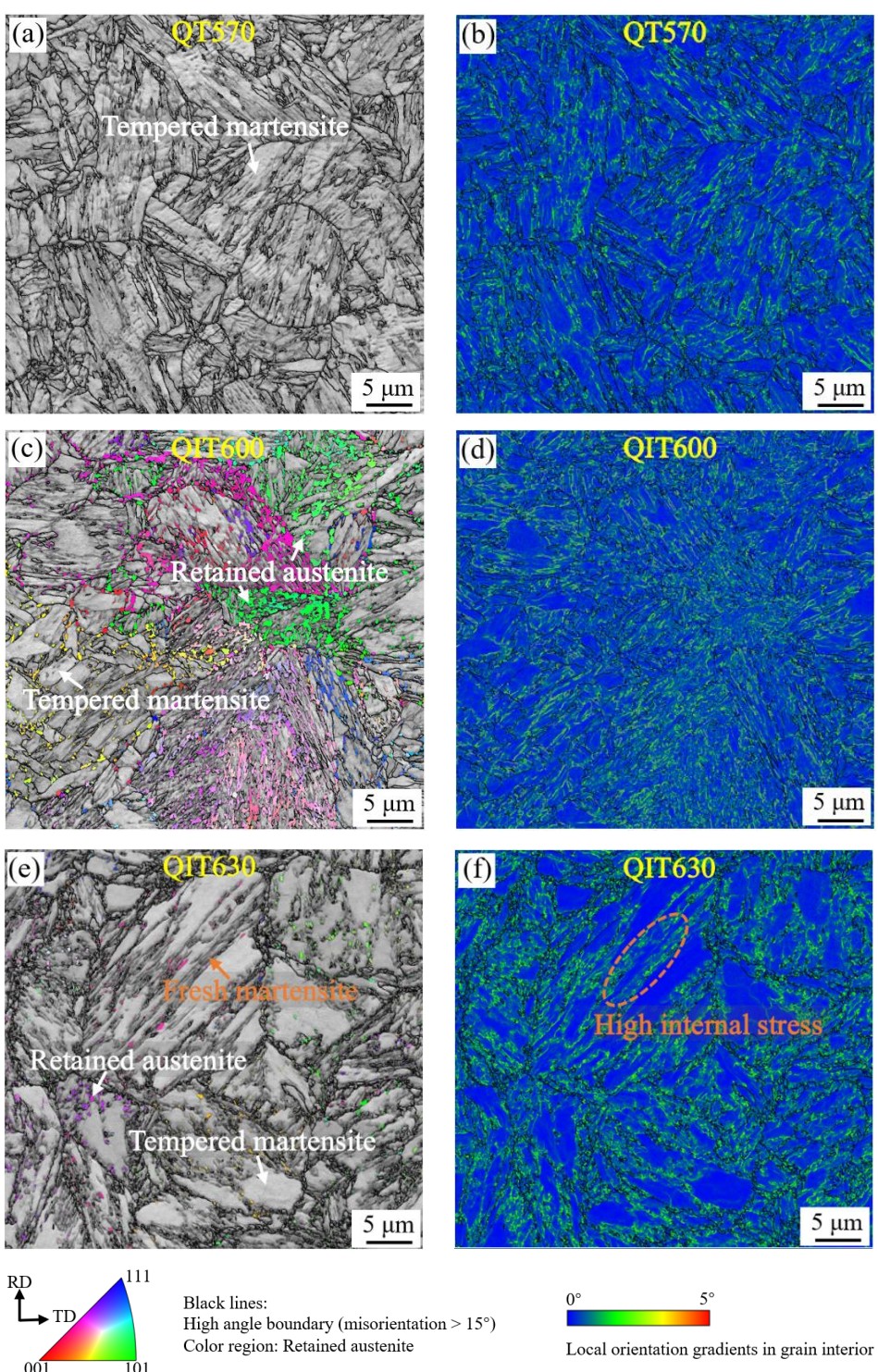

**Figure 6.** EBSD crystallographic analyses of the (**a**,**b**) QT570, (**c**,**d**) QIT600, and (**e**,**f**) QIT630 steels; (**a**,**c**,**e**) EBSD image quality—IPF of retained austenite; (**b**,**d**,**f**) KAM of matrix—retained austenite/fresh martensite phase mappings.

The three tempering temperatures (570, 600, and 630 °C) were used to regulate the microstructures of the low-carbon alloy steel. Compared with the QT570, 18 vol.% and

1.5 vol.% retained austenite formed in the QIT600 and QIT630 steels, respectively. The key to increasing the content of retained austenite is to regulate the tempering temperature in order to form stable reversed austenite. In addition, regulating the tempering temperature refines the size of VC precipitates.

### 3.2. The Mechanical Properties of the Material

The stress–strain curves and work-hardening rates of QT570, QIT600, and QIT630 are displayed in Figure 7a,b. The QT570 steel has a yield strength of 1150 MPa, an ultimate tensile strength of 1170 MPa, and a total elongation of 16.5%, indicating strong mechanical properties. The high yield strength and yield ratio of 0.98 in QT570 steel can be attributed to the fine martensite laths with a high density of dislocations. Additionally, the tensile stress–strain plots for QT570 steel display a clear yield plateau that forms and spreads under constant stress and does not undergo work hardening. This plateau is influenced by the strain-hardening ability of the test steel and has a significant impact on its mechanical properties [20]. Even after being tempered at 570 °C, QT570 steel retains a high density of dislocations, resulting in a more pronounced yield plateau during tensile testing due to the presence of numerous tangled dislocations that impede the movement of mobile dislocations.

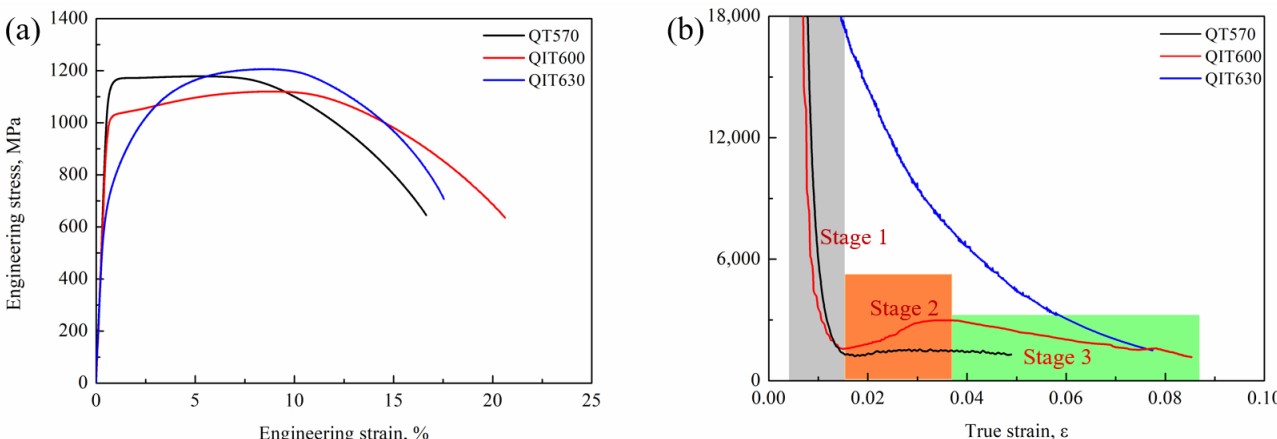

**Figure 7.** Tensile properties: (**a**) engineering stress–strain curves; (**b**) work-hardening rate–true strain curves.

For QIT600 steel, the yield strength decreases to 1025 MPa, the ultimate tensile strength decreases to 1120 MPa, the yield ratio decreases to 0.92, and the total elongation increases to 21%. According to the intercritical tempering transformation process, the volume fraction of the soft retained austenite of QIT600 steel is significantly higher than that of QT570 steel, which is the main factor leading to the lower yield strength of QIT600 steel than that of QT570 steel. In addition, the softening of martensite laths and a lower dislocation density also reduce the yield strength of QIT600. However, the TRIP effect of this phase enhances the work-hardening capability, which leads to an increase in the tensile strength. Consequently, the yield ratio was significantly lower. In addition, the QIT600 steel affords a continuous work-hardening ability. As a result, it shows continuous yield behavior due to the TRIP effect of the widely distributed retained austenite.

During the tensile process of the QIT600 steel, the work-hardening rate exhibited three stages: a sharp decrease, a slight increase, and a slow decrease (Figure 7b). In stage 1, the work-hardening rate decreased significantly with strain, mainly associated with the dynamic recovery of dislocations [21]. In stage 2, the work-hardening rate increased and reached a peak value because of the transformation of a large fraction of retained austenite into martensite, and this high-mechanical-stability retained austenite remained after stage 1 deformation. In stage 3, the work-hardening rate gradually decreased because the TRIP could not maintain a high work-hardening capacity. In addition, the QIT600 steel

exhibited a remarkable work-hardening rate due to the continuous TRIP effect, which is attributed to the 18% volume fraction of retained austenite (Figure 7b). Dynamic strain partitioning based on the TRIP effect is beneficial for subsequent homogeneous deformation; consequently, ductility was significantly increased.

The QIT630 steel displays a yield strength of 716 MPa, an ultimate tensile strength of 1206 MPa, and a total elongation of 17.5%. Although QIT630 steel has a lower yield strength compared to the other two steels, its ultimate tensile strength surpasses that of the others because of its microstructure mainly consisting of tempered martensite and fresh martensite. Tempered martensite is a soft phase, resulting in the low yield strength of QIT630. Nonetheless, through the high-temperature isothermal intercritical tempering process, fresh martensite is generated from alloy-enriched austenite. This newly formed martensite has a high density of dislocations and hardness, resulting in a significant increase in ultimate tensile strength [22]. The volume fraction of fresh martensite must be controlled in multiphase steels as it has been reported to contribute the most to ultimate tensile strength [23]. In addition, the yield ratio of QIT630 steel is significantly reduced to 0.59 because of the substantial difference in strength between tempered martensite and fresh martensite.

Figure 8 displays the impact energies of the tested steels at temperatures of −80 and −196 °C. Specifically, the QIT600 steel exhibited impact energy of 1.88 MJ/m$^2$ at a temperature of −80 °C, while the QT570 and QIT630 steels demonstrated impact energies of 1.06 and 1.19 MJ/m$^2$, respectively. At −196 °C, the impact energy of QIT600 steel remained consistently high at 1.25 MJ/m$^2$. In contrast, the impact energies of QT570 and QIT630 sharply decreased to 0.23 and 0.61 MJ/m$^2$, respectively. Therefore, the QIT600 exhibited superior low-temperature impact toughness compared to the other tested steels due to its higher volume fraction of retained austenite.

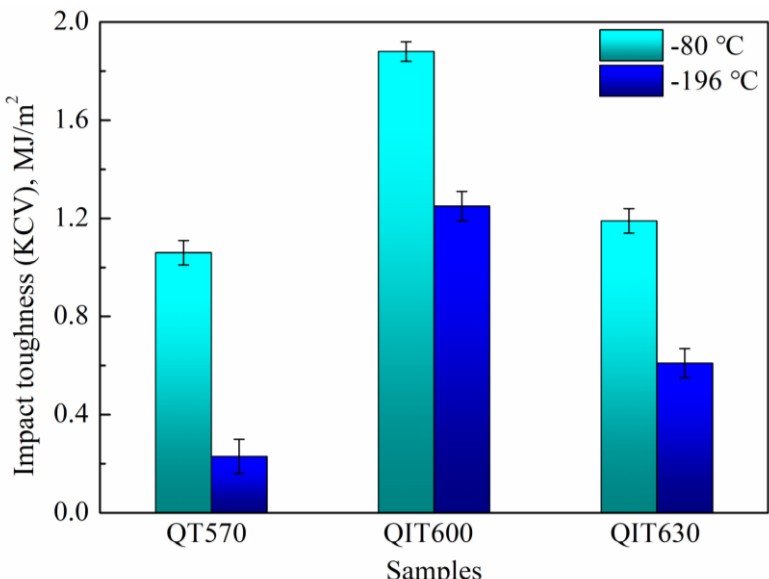

**Figure 8.** Impact energy at −80 and −196 °C.

Previous studies have suggested that the presence of retained austenite can effectively absorb a significant amount of dislocations from neighboring martensite laths, thereby contributing to an improvement in toughness [24]. Furthermore, the presence of retained austenite is expected to hinder crack initiation and propagation, effectively enhancing toughness [25]. A surface analysis was conducted to elucidate the toughening mechanism of the tested materials and discuss their fracture modes. The SEM graphs depicting the impact fractures of the QT570, QIT600, and QIT630 steels at −196 °C are presented in Figure 9. Fractography analysis of the cryogenic impact experiments revealed that QT570 steel displayed numerous cleavage facets, secondary cracks, and intracrystalline fractures

characteristic of brittle fracture, as shown in Figure 9a. On the other hand, QIT600 steel showed large dimples with surrounding fine dimples on its fracture surface (Figure 9b), suggesting a high plastic strain capacity at low temperatures. The fracture morphology of QIT630 steel demonstrates the presence of a small quasi-cleavage cracking facet. The fracture surface exhibits a short river pattern adorned by tear ridges, along with minor secondary cracks and fragments of small indentations, as shown in Figure 9c. These features suggest a quasi-cleavage fracture characteristic.

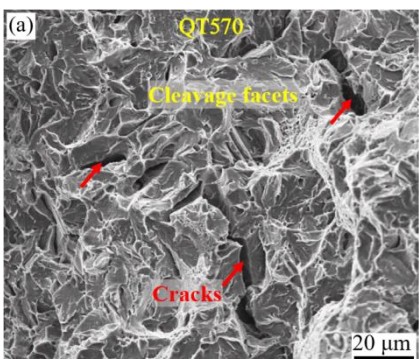 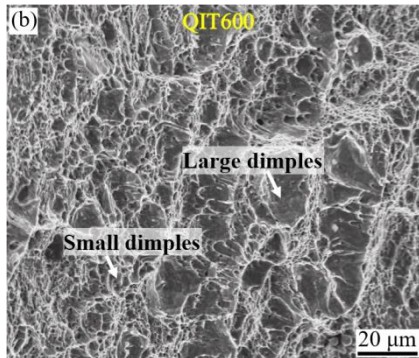 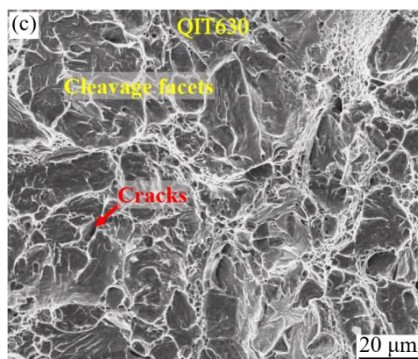

**Figure 9.** Impact fracture micrographs at −196 °C: (**a**) QT570; (**b**) QIT600; (**c**) QIT630.

Strip-like M–A constituents are distributed in the tempered martensite matrix of QT570 steel. These hard and brittle M–A constituents may impede dislocation movement, create dislocation pile-ups, and generate additional internal stress because of the deformation mismatch with the surrounding matrix. The interaction between dislocation pile-ups and additional internal stress can lead to stress concentration. When the stress concentration exceeds the strength of the M–A constituents or the interface bonding strength, the M–A constituents may crack or separate from the matrix interface, resulting in microcracks, cleavage fracture, and deteriorating steel toughness [26,27]. Therefore, the presence of M–A constituents leads to a reduction in impact energy due to cleavage cracking. However, for QIT600 steel, a significant amount of retained austenite can greatly enhance its low-temperature toughness. Previous research has shown that the conversion of retained austenite can improve impact toughness by reducing local stress concentration, increasing plastic strain capacity, delaying crack initiation, and impeding microcrack propagation [28,29]. QIT600 steel has the highest volume fraction of metastable retained austenite, resulting in a significant improvement in toughness due to transformation-induced effects. The continuous transformation of retained austenite during plastic deformation at low temperatures results in significant TRIP toughness and deflects crack propagation, leading to improved toughening [30]. Therefore, the TRIP effect of retained austenite mainly enhances crack propagation energy during impact loading. On the other hand, the poor toughness of QIT630 steel is attributed to the formation of fresh martensite within its tempered martensite matrix. Hard fresh martensite serves as the primary site for crack initiation and propagation, resulting in a reduction in low-temperature toughness [31]. Therefore, the presence of newly formed martensite in QIT630 steel leads to a reduction in its low-temperature toughness.

## 4. Conclusions

This study examined the impact of intercritical tempering on both the mechanical properties and microstructure of high-strength low-carbon steel. The conclusions are summarized as follows:

(1) In terms of microstructure, QT570 steel primarily consisted of a tempered martensite matrix. On the other hand, QIT600 steel exhibited a mixed microstructure comprising mainly tempered martensite along with 18 vol.% retained austenite. The phases present in QIT630 steel consisted of tempered martensite and fresh martensite. Addi-

tionally, the size of the VC precipitates in the QIT600 steel was smaller compared to that in the QIT630 steel, suggesting a higher precipitation-strengthening effect.

(2) The retained austenite in QIT600 steel exists in block-like and strip-like forms, with the block-like retained austenite having an alloy element composition of 12.15Ni-4.7Mn-1.96Cr and the strip-like retained austenite having a composition of 13.64Ni-4.84Mn-2.44Cr. Both types of retained austenite are enriched with high-concentration alloy elements, which ensure their thermal stability at −196 °C.

(3) Excellent combination of high yield stress (1025 MPa), perfect total elongation (21%), and excellent cryogenic impact energy (1.25 $MJ/m^2$ at −196 °C) was achieved in the QIT600, surpassing the strength–ductility–toughness combination of the QT570 and QIT630 steels with their respective yield strengths of 1150 MPa and 716 MPa, total elongations of 16.5% and 17.5%, and cryogenic impact energies of only 0.23 and 0.61 $MJ/m^2$.

(4) The mechanical stability of retained austenite in QIT600 steel enables the occurrence of TRIP over a wide range of strains. This leads to sustained work hardening and a significant improvement in ductility. The TRIP toughening effect produced by the retained austenite effectively enhances the ability to arrest cracks during low-temperature impact deformation, ultimately resulting in a ductile fracture behavior.

**Author Contributions:** Conceptualization, L.S.; methodology, L.S.; validation, L.S. and J.Z.; formal analysis, L.S. and F.W.; investigation, L.S., J.W., J.Z. and F.W.; resources, G.Y.; data curation, F.W.; writing—original draft preparation, L.S.; writing—review and editing, J.W. and J.Z.; visualization, L.S. and J.W.; supervision, G.Y. and G.W.; project administration, G.Y. and G.W. All authors have read and agreed to the published version of the manuscript.

**Funding:** This research received no external funding.

**Data Availability Statement:** Not applicable.

**Conflicts of Interest:** The authors declare no conflict of interest.

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
