# Peer review of "The Influence of Tempering Temperature on Retained Austenite and Ductility–Toughness of a High-Strength Low-Carbon Alloyed Steel"

_crystals, doi:10.3390/cryst13081194_

Round 1
Reviewer 1 Report (Previous Reviewer 2)
Comments and Suggestions for Authors
The article can be published after correcting the comments made.

Author Response
Please see the attachment

Reviewer 2 Report (New Reviewer)
Comments and Suggestions for Authors
The work studies the effect of tempering temperature on high-strength, low-carbon alloy steel.
Presents microstructural characterization (SEM, TEM, and EBSD) quantification of austenite retained by XRD and mechanical properties consistent with the stated objective.
Minor comments:
Abstract: To clarify, it suggests that they are types of steel, but it is a composition with three tempering temperatures. This should be redrafted.
Remember to include the axis values in Figure 1.
Remove the yellow highlighting on the paragraph.

Author Response
Please see the attachment

This manuscript is a resubmission of an earlier submission. The following is a list of the peer review reports and author responses from that submission.
Round 1
Reviewer 1 Report
Comments and Suggestions for Authors
Selected points needing an improvement:
Use the same name for the studied steel throughout the paper: first in full and then an abbreviation.
Ac1 and Ac3 temperatures can be also indicated on Figure 1 for more convenience.
Please discuss why those three tempering temperatures (570, 600, and 630 ℃) were used.
Please explain why this specific 2θ range was used (45-120 °). Also, the XRD detection limit should be mentioned in the text.
The quality of the images presented on Figure 2 is not sufficient: these look overetched and out of focus. Tempered martensite morphology should be shown at higher magnification to be able to distinguish it from the fresh martensite. It is also necessary to explain how the retained austenite was defined on these images. Finally, it is necessary to add some details to the description of the thermal treatments as to explain the presence of the fresh martensite.
Figure 3: Please explain why there is the (311) peak for austenite in QT570 sample, but the austenite volume fraction is zero. Also, it seems important to have a full XRD spectrum presented in the paper. Finally, the measurement for the as quenched sample should be presented.
More details are needed for the TEM images presented in Figure 4 and 5: what type of imaging was used. Also, the resolution is not sufficient in Fig 4 a-c: it is not possible to distinguish between the tempered martensite morphology. In Figure 5e, the retained austenite does not look like films. It has the same morphology as in Figure 5b, but smaller in size. This needs to be clarified.
The presentation of the Figure 6 showing EBSD results should be improved; all the colours and misorientation maps should be explained in the legend More explanations should be given for Figure 6 b, d, f in both text and legend.
The section on mechanical properties should be improved as to use proper names in English for different terms, e.g. yield plateau, continuous yielding etc. There is a mistake in the legend for the Figure 7.
Please explain why MA constituents were not showed prior to line #298.
Sections between lines 347 and 375 are not filled in.
Comments on the Quality of English LanguageAn extensive English editing is necessary for the presented paper.
Author Response
Thank you for your comments.I have made the following changes in response to your comments

Reviewer 2 Report
Comments and Suggestions for Authors
The article can be published after correcting the comments made in the review.

Minor editing of English language required.
Author Response
Thank you for your comment.I have made the following changes in response to your comments

Round 2
Reviewer 1 Report
Comments and Suggestions for Authors
I have carefully examined author’s replies to my comments and took the decision to reject the paper in the present form. The motivation is that the author’s analysis of the XRD results and microstructural examinations (SEM, TEM) are not sufficient for an archival journal paper. It is suggested to take time to perform additional experiments and to carefully analyse and interpret the obtained results. It is also highly recommended to obtain a professional English editing to improve paper clarity.
Comments on the Quality of English LanguageIt is highly recommended to obtain a professional English editing to improve paper clarity.